# Ontological Analysis of COVID-19 Vaccine Roll out Strategies: A Comparison of India and the United States of America

**DOI:** 10.3390/ijerph18147483

**Published:** 2021-07-13

**Authors:** S. D. Sreeganga, Ajay Chandra, Arkalgud Ramaprasad

**Affiliations:** 1Ramaiah Public Policy Center, Bengaluru, Karnataka 560054, India; ajay.chandra@rppc.ac.in (A.C.); arkalgud.ramaprasad@rppc.ac.in (A.R.); 2Information and Decision Sciences Department, University of Illinois at Chicago, Chicago, IL 60607, USA

**Keywords:** COVID-19 vaccine, vaccine roll out, national strategies, ontology, COVID-19

## Abstract

The unprecedented outbreak of the COVID-19 pandemic has forced governments to devise national strategies to curtail its spread. The present study analyzes the national strategies of India and the United States for the COVID-19 vaccine roll out. The paper presents an ontology of COVID-19 vaccine roll out, maps the national strategies, identifies, analyzes the emphases and gaps in them, and proposes corrections to the same. The analysis shows that the national strategies are selective in their focus and siloed in their approach. They must be systematized to address the emerging challenges effectively. Thus, there is need for a systemic understanding and analysis to reinforce the effective pathways to manage vaccine roll out, reposition the ineffective ones, and engineer new ones through feedback and learning.

## 1. Introduction

Since the COVID-19 pandemic started around the end of 2019, the main strategies to contain and control the virus were interventions such as social distancing, lockdown, and wearing personal protective equipment [1]. To tackle the pandemic on the scientific front, scientists across the world joined hands for innovative tie-ups with both the pharmaceutical industry and the medical sector. They collaborated in the development of treatment protocols and of vaccines to impede the progress of this overwhelming pandemic [2]. The fastest time period any vaccine had previously been developed was 4 years, for mumps in the 1960s [3]. The scientists were able to develop COVID-19 vaccines swiftly because of years of previous research related to viruses, faster manufacturing capacity, and large funding [4]. In addition, regulators’ quick approval of the vaccines for emergency use allowed countries to start their COVID-19 vaccine roll out strategies [1].

Even as many nations rapidly scaled vaccine development [5], the expedited approval of the COVID-19 vaccine initially created considerable vaccine-related fears about their efficacy and safety. In addition, countries faced challenges related to distribution, demand, supply, and providing equitable access to people [6]. Though vaccination strategies of the U.S., India, and European countries had similar plans to prioritize recipient groups, their distribution roll out strategies were different. The differences are in terms of start date of vaccinating the prioritized population, rate of vaccination, and strategies related to the timing of the doses, etc. [7]. For the COVID-19 mass vaccination program, countries have faced varied roadblocks due to large demography, capacity of the health infrastructure, lack of sustainable investments, vaccine hesitancy, adequate supply, and distribution-related challenges [6,8]. Assuming vaccination is feasible, the strategy for its introduction must consider the structure of the health system and strength of the local healthcare system, service funding, and monitoring of success and failures for feedback and learning [9]. In such scenarios it becomes essential for countries to pay attention to factors such as manufacturing complexities, given the variations in demand, production capacity in terms of cost, unit set-up, labor, regulatory issues, stockpile, funding avenues, decentralization for administering the vaccine, and science communication [10].

Due to the above complexities, there is a need for countries to have a supportive framework for implementation of vaccine strategies [9]. The framework should be such that it represents the system complexity systematically and symmetrically. The paper uses an ontological framework to do so. With countries having unique challenges in their vaccine roll out, the pandemic continues to grow. Different countries have prepared national-level strategies to guide and navigate through the vaccine roll-out process. It is imperative at this point to have a global view of the strategic actions carried out since the COVID-19 vaccination movement started, to assess, learn, apply, and re-learn from the process. 

The paper analyzes the national strategic documents of two countries, India, and the U.S., through the lens of an ontology (Figure 1). For India, the “COVID-19 Vaccines Operational Guidelines” by the Ministry of Health and Family Welfare (MoHFW), Government of India (GoI) [11] and for the United States (U.S.) the “National Strategy for the COVID-19 Response and Pandemic Preparedness” by the office of President Joseph R. Biden, Jr. [12] are used for analysis. The aim is to analyze both the strategic documents and map the sections of the documents onto the ontological framework. The analysis of these countries gives a comparative view of the strategic actions of a developing and a developed country. This will help to visualize the various similarities and differences in the barriers to and drivers of various vaccination strategies/actions and how such countries can continue to fight the pandemic.

## 2. Materials and Methods

### 2.1. Ontology of COVID-19 Vaccine Roll Out

The ontology of the COVID-19 vaccine roll out defines its dimensions, elements, and boundaries [13]. It deconstructs the policy problem’s complexity hierarchically [14], visualizes it in structured natural English, and encapsulates its combinatorial logic [15]. It organizes the terminologies, taxonomies, and narratives of the policy problem systemically, systematically, and symmetrically [15,16,17,18,19]. It is a cognitive map of the system [20,21,22] to (a) design the policy alternatives, (b) determine effective, ineffective, and innovative policies, and (c) direct the choice through feedback and learning [23,24]. It is a qualitative theory [25] of the policy problem.

Similar ontologies have been used to conceptualize and analyze mHealth [16] , health care systems [26], learning surveillance systems [27], and higher education policies [28]. The development and application of the ontology in the current paper follows the description of the logic and process by [13]. 

The ontology is shown in Figure 1. The actions for vaccine roll out are listed in the middle column of the ontology. They are to educate, prioritize, locate, communicate, coordinate, mobilize, and redirect. The resources for COVID-19 vaccination are spatial (distance and location), temporal (availability and scheduling), financial (income and expenditure), informational (stimulant and educational), human (psychological, sociological, cultural, and other), and technological (IT, transportation, and medical). The resources are listed in the second column from the left in the ontology. Allocation of these resources is at different levels. The levels are national, state, district, and block, and are listed in the first column of the ontology. The forces affecting vaccination roll out can be barriers to it, inhibitors of it, catalysts of it, and drivers of it. They are listed in the third column from the left. The vaccination is processed through different stakeholders. They are listed in the rightmost column of the framework under Stakeholder. The recipients of the vaccine are different groups listed under Population in the second column from the right. It includes: (a) priority groups—of workers (health and frontline workers), by age, by geography, and other special needs, and (b) the general population. A glossary of the dimensions and elements of the ontology is given in Appendix A.

The ontology encapsulates 4 × 15 × 4 × 8 × 6 × 19 = 218,880 systemic transverse pathways that can be harnessed to address the issue of vaccine roll out systematically. (It is the product of the number of base-level elements in each column of the ontology.) Each pathway is a concatenation of an element from each column (dimension) of the ontology together with the adjacent words/phrases. Two illustrative pathways are: National level spatial distance barrier for vaccine roll out to educate priority group workers health for COVID-19 vaccination by national government.Block level technological medical driver for vaccine rollout to redirect general population for COVID-19 vaccination by NGO.

Some of the pathways may be effective, others ineffective, and many infeasible. Their efficacy must be discovered through research and practice. By the same token, some pathways may be known, others overlooked, and many unknown and to be discovered.

### 2.2. Method

The paper analyzes the strategies and guidelines issued by India (“COVID-19 Vaccines Operational Guidelines”) by the Government of India (GoI) [11] and the U.S. (“National Strategy for the COVID-19 Response and Pandemic Preparedness”) by the office of President Joseph R. Biden, Jr. [12] for COVID-19 vaccination by mapping them onto the ontology. The mapping is then used to generate the monads map and themes map to visualize the landscape of the domain—the relative emphasis on the different elements and themes. Subsequently, it discusses the implications of the emphases and the gaps in them—individually and in comparison with each other—for effective and efficient vaccine roll out in the respective countries.

### 2.3. Mapping

The inclusion and exclusion criteria that were used for coding is summarized in Table 1. Each strategy or action point was jointly reviewed to identify the reference to each ontological element. The elements present in each strategy were marked in a custom-designed Excel spreadsheet. The mapping is binary—either present (1) or absent (0)—and not weighted. Only the dimensions and elements explicitly articulated in the strategy or action points were mapped. The mapping of both the documents went through two iterations by two of the authors to ensure its reliability and validity. After the rounds of individual mapping, the mappers discussed the discrepancies in their mapping and arrived at a consensus for the final mapping. The glossary in Appendix A was used to assure the validity of the mapping.

## 3. Results

The monads maps in Figure 2 and Figure 3 numerically and visually summarize the frequency of occurrence of each dimension and element of the ontology in the Indian and U.S. documents, respectively. The number adjacent to the dimension name and the element is the frequency of occurrence across the various strategies/actions mentioned in the respective document. The bar below each element is proportional to the frequency relative to the maximum frequency among all elements. Since each item can be coded to multiple elements of a dimension, the sum of the frequency of occurrence of elements may exceed the frequency of occurrence of the dimension to which the elements belong. The Indian and U.S. monads maps are described below.

### 3.1. India’s Monads Map

The dominant focus of the strategies is on the resources (84), action (84), and force (60). There is substantial focus on the stakeholders involved (56) and the level (48). There is less focus on the population type (25).

Amongst the action elements, the dominant focus is to manage (65). There is some emphasis to educate (35), mobilize (34), redirect (33), coordinate (32), communicate (30), and prioritize (29). The least emphasis is on locate (2).

The strategies mentioned in the document cover a spectrum of resources for the vaccine roll out. It is heavily focused on the informational stimulant (48), technological IT (44), spatial location (42), and temporal availability (38). There is medium focus on technological transportation (34), human other (32), informational education (31), and technological medical (29). There is some emphasis on temporal scheduling (20), spatial distance (10), and financial expenditure (10). The least-focused resources are human sociological (6), financial income (3), human cultural (1), and human psychological (1). 

A significant proportion of strategies consider the forces that affect the actions. The most focus is on the drivers (45) for vaccine roll out; there is lesser emphasis on the catalysts (13). There is little emphasis on inhibitors (7) and least on the barriers (5).

The levels at which the actions are to be executed are focused predominantly on the district (32) and state (31) levels. There is moderate emphasis on national (20) and block (19) levels.

Among the stakeholders, the majority focus is action by the government district (22), health provider frontline worker (21), health provider physician (19), government state (18) and agent hospital (17). There is medium focus on health provider field worker (14), agent other (14), government local (12), agent NGO (12), health provider nurse (11), agent community (10) and government national (10). Among the less focused are health provider volunteer (9), health provider pharmacist (7), agent health center (7), and agent clinic (2). Elements such as government international, agent pharmacy, and agent family have not been emphasized.

Amongst the population, greater emphasis is on priority workers health (16) and priority workers frontline (16). Compared to targeted essential workers, age-based vaccine roll out is less effective and less equitable [1]. However, the MoHFW document has placed medium emphasis on priority group age (13) and priority group geographic (10). Priority group special (4) and general population (4) received lesser emphasis Figure 3.

### 3.2. The U.S. Monads Map

The results of mapping the corpus of the U.S. National Strategy for COVID-19 Response and Pandemic Preparedness onto the ontology are presented in the monads map in Figure 3. A total of 94 items of the U.S. National Strategy for the COVID-19 Response and Pandemic Preparedness were mapped. 

The dominant focus of the strategies is on resources (88), forces (84), type of action (80), and the stakeholders involved (76). There is less focus on the level (30) and even less on the population type (19).

Although all 94 items are linked to the strategies, only 80 refer to the type of strategy to be used. The dominant focus is to manage (46) and the next is to mobilize (37). There is lesser emphasis on actions such as communicate (18), coordinate (17), educate (15), prioritize (14), redirect (9), and locate (3). 

The strategies laid out in the document cover a spectrum of resources for the vaccine roll out. It is heavily focused on the informational stimulant (42) and the temporal availability (33). There is medium focus on technological medical (26), technological transport (24), and informational education (20). There is some emphasis on spatial location (18), human other (17), technological IT (16), financial expenditure (12), and temporal scheduling (10). The resources least focused are human sociological (5), financial income (5), human cultural (3), and spatial distance (1). There is no mention of human psychological (0) resources. 

The levels at which the actions are to be executed are focused predominantly on the state level (23). Next, there is emphasis on block (18) and district (17), and last on national levels (15).

A significant proportion of strategies consider the forces that affect the actions. The most focus is on the catalyst (48); there is lesser emphasis on the drivers (35). There is little emphasis on inhibitors (11) and least on the barriers (8).

Among the stakeholders, the majority focus is action by the national government (56). There is medium focus on agent community (20), agent other (18), and state government (18). Among the less focused are health provider nurse (13), district and local government (12), health provider physician (11), health provider frontline worker (10), health provider pharmacist (8), agent clinic (8), agent health center (8), health provider field worker (7), health provider volunteer (6), agent hospital (5), agent pharmacy (5), agent NGO (5), and government international (2). There is no mention of agent family (0). 

The strategy document focuses least on the population (19). The dominant focus is the priority group age (8) and geographic (8). The next most emphasized are priority group special (7), priority group health workers (6), and general population (6), and last is the priority group frontline worker (4).

### 3.3. Theme Maps

The theme maps (Figure 4 and Figure 5) visually summarize the co-occurrence of elements of the ontology in the Indian and U.S. vaccine rollout strategies, respectively. Hierarchical cluster analysis was done using SPSS (Statistical Package for Social Sciences; IBM: Chicago, IL, U.S.) with simple matching coefficient (SMC) as the distance measure and the nearest-neighbor aggregation procedure (it groups similar elements/objects into clusters). SMC considers both presence (coded “1”) and absence (coded “0”) elements equally. The detailed rationale for the choice of the clustering method and the presentation of the results is given in Syn and Ramaprasad and La Paz et al. [29,30]. The five themes represent the five equidistant clusters in the dendrogram of the agglomeration [29]. The colors in the figures highlight the elements of the five themes.

#### 3.3.1. India’s Themes Map

Thematically, the primary theme (in red) represents informational stimulant to manage/redirect the vaccine roll out. The secondary theme (in brown) represents the combination of the following resources to educate and communicate for the vaccine roll out: spatial location, temporal availability, informational educational, human other, and technological IT. The tertiary theme (in yellow) represents the combination of temporal scheduling, and technological transportation/medical resources at all levels to prioritize, coordinate, and mobilize the vaccine roll out by the government at state and district levels. The quaternary theme (in blue) represents inhibitors and catalysts of vaccine roll out in government (national, local), healthcare providers (physicians, frontline workers, field workers, and volunteers), and agents (hospitals, NGOs, and others). The quinary theme represents the absence of the population elements, some healthcare providers (nurses, pharmacists), many agents (clinics, health centers, pharmacies, families, and communities), barriers to vaccine roll out, location of action, spatial distance, financial income and expenditure, and human psychological, social, and cultural elements in the themes. 

#### 3.3.2. The U.S. Themes Map

The colors in Figure 5 highlight the five themes in the U.S. National Strategy for the COVID-19 Response and Pandemic Preparedness. The primary theme in red, is catalyst for vaccine roll out by the national government. The secondary theme in brown is the combination of temporal availability and informational stimulant drivers to manage the vaccine roll out. The tertiary theme in yellow is the combination of spatial location and human resources (other) to mobilize the vaccine roll out. The quaternary theme in blue is national level combination of temporal scheduling, financial expenditure, information for education, and technology (IT, transportation, medical) to address inhibitors of vaccine roll out through education, prioritization, communication, coordination, and redirection by community and other agents. The quinary theme represents the absence of the population elements, most of the stakeholders, barriers, action location, levels below the national level, and many resources (spatial distance, financial income, human—psychological, sociological, and cultural) in the themes.

Overall, the themes are in order of decreasing dominance in the corpus of the Indian and U.S. national documents—the primary theme is the most emphasized and the quinary theme is absent. The themes are segmented—they span only a few of the six dimensions of the ontology. India’s primary and secondary themes span over two dimensions each, while the U.S. primary theme spans over two and the secondary theme spans over three dimensions. India’s tertiary themes span over four dimensions, and the U.S. tertiary theme spans over two. Lastly, India’s quaternary theme spans across two dimensions and the U.S. quaternary theme spans across five dimensions. All the Population elements, most Stakeholder elements, most Level elements, and a few Resource, Force, and Action elements are excluded. The themes are skewed. None of the themes includes barriers. Overall, the coverage of the Indian and U.S. National Strategy is not systemic but is fragmented.

## 4. Discussion

Given the rapid outbreak of COVID-19 and its increasing prevalence, the need to devise timely measures for vaccine roll out has been critical. India’s central guidelines for vaccination rollout and the U.S. National Strategy for the COVID-19 Response and Pandemic Preparedness were devised to tackle this ongoing outbreak. The following section discusses the emphases and gaps in each and in comparison with each other.

### 4.1. India’s COVID-19 Vaccines Operational Guidelines—Emphases and Gaps

The heavy emphases on resources and actions are understandable and appropriate. They are the main axes of any successful vaccine roll out. 

The heavy emphasis on drivers/catalysts as compared to that on barriers/inhibitors suggests a skewed view of the ease of vaccine roll out. Policy options such as subsidies for basic vaccine research, liability protection for manufacturers, and fast-track approval for new vaccines [10] are all critical for temporal availability of vaccines. The guidelines appear to be blind to the characteristics of different populations, their (un)willingness to be vaccinated, and their resistance to it. The population groups differ in their risk profiles, vaccine hesitancy, numbers, and many other parameters. These could pose barriers to and inhibit the proposed actions. Ideally, they should be differentiated and integrated. 

Amongst the resources, the emphases on the informational stimulant, technological IT, spatial location, and temporal availability stand to reason. The lower emphases on transportation technology, other human resources, information for education, medical technology, and time scheduling could perhaps be explained. However, the very low emphases on spatial distance, financial expenditure, human sociology, financial income, human culture, and human psychology resources appear to be a significant oversight. These last set of elements have been the major barriers to, and inhibitors of vaccine roll out. Human psychology and associated decision making are critical for tackling issues pertaining to vaccine hesitancy. In addition, due to lower emphases on information for education, there is also a paucity of studies to evaluate the willingness of people, facilitators and barriers in vaccine roll out [31] in India.

While the guidelines cover a wide range of stakeholders, the low-to-no emphases on clinics, pharmacies, and families is an oversight. The clinics have emerged as an important link in the vaccine roll out. States have now devised detailed plans for clinics to intimate information about the weekly doses available and to build confidence that enough doses are in the pipeline for people to avail their second dose [32].

Thematically, the emphases and gaps highlight the segmented, selective, and siloed logic of the guidelines. While the guidelines cover most of the elements of the framework (and hence the system) as shown in the monads map (Figure 2) the coverage is not systematic. The primary theme to use informational stimulant as a driver to manage and redirect vaccine roll out is not linked to any level of the government, target population or stakeholder. 

Similarly, the secondary theme to use spatial-location, temporal availability, information for education, other human, and technological IT resources to educate and communicate is also not linked to any level of the government, population, or stakeholder. Temporal availability of the existing vaccines facilitates reduction in risk of illness, prevent infection and onward transmission [33]. Access to more vaccines, and multilateral contracts between the government should aid in enhancing the speed, coverage, and availability of vaccine rollout. Reports of violations of informed consent in trials and inadequate transparency around adverse events following immunization (AEFIs) [34] exist; the need to disseminate appropriate educational information is critical to ensure vaccine uptake in India.

The tertiary theme is the least segmented and covers four of the six dimensions. It too excludes the population elements and the forces. It integrates all the levels of the government with the state and district government stakeholders to use time scheduling, transportation technology, and medical technology (transport and storage logistics, cold-chains supply [35]) to prioritize, coordinate, and mobilize the population for the vaccine roll out. Strong support from government, healthcare organizations, culturally appropriate local approaches [9], NGOs, and public—private partnerships are necessary for optimizing vaccine outcomes. Global shortage of vaccines is inevitable, and given the magnitude of the task of scaling up production; it is necessary to prioritize groups [33] across the national, state, district, and local levels. 

The quaternary theme highlights the inhibition and catalysis by a variety of stakeholders, but not with reference to any population, action, resource, or level. The importance of coordinating delivery mechanisms, timely deployment of a skilled cadre of health workers, and the need to reach out to disadvantaged groups [33] were some of the insights found during the vaccination campaign in Israel. The need for a coordinated response across multiple levels of the health system is critical to mitigate [36] subsequent COVID-19 hits. India should use multipronged strategies such as special campaigns, localized communications, door-to-door outreach campaigns, text messaging of immunization campaigns, and implementing affordable or free immunization visits in order to sustain gains in vaccination coverage [37]. 

The quinary theme highlights very significant oversights. Three examples are: (a) human psychological, sociological, and cultural barriers to locate people for vaccine roll out in different populations, (b) financial income and expenditure barriers to locate people for vaccine roll out in different populations, and (c) the absence of nurses and pharmacists in the vaccine roll out. It was observed in Israel that effectiveness of mass vaccination and relative risk reduction was mainly due to the measures of vaccine coverage and allocation to different groups [38]. Vaccine strategies that are considerate of diverse populations and risk profiles of groups /individuals are critical. For instance, early strategies and measures to vaccinate essential workers could result in substantial reductions in the number of infections, hospitalizations, deaths, and COVID cases [1]. 

### 4.2. The U.S. National Strategy for the COVID-19 Response and Pandemic Preparedness—Emphases and Gaps

The heavy emphases on resources, actions, and stakeholders are understandable and appropriate. They are the critical axes of any successful vaccine roll out.

As in the case of India, the heavy emphasis on drivers/catalysts as compared to that on barriers/inhibitors suggests a skewed view of the ease of vaccine roll out. The guidelines appear to be blind to the characteristics of different populations, their (un)willingness to be vaccinated, and their resistance to it. The strategies appear to have failed to anticipate the importance of prioritizing these groups with different characteristics and differentiating among them. For example, strategies for population segments need not only be for and according to demographic characteristics such as age, but need to address those that are barriers created by self-identity, social beliefs, or by in-groups [39]. These could pose barriers to and inhibit the proposed actions. Thus, the guidelines must ideally have differentiated and integrated target populations. 

Amongst the resources, there is heavy emphasis on the informational stimulant; that is followed by temporal availability, transportation technology, medical technology, and educational information. All these could be catalysts and drivers of actions. On the other hand, spatial distance could be a significant barrier to vaccine roll out, especially in rural areas. For instance, a study of disparities in COVID-19 vaccination coverage between urban and rural counties in the United States suggests that due to the challenges with vaccine access and the dearth of pharmacies in rural areas, people have had to travel to nonadjacent counties [40]. Financial income and expenditure also could be a significant force, positive or negative, in the vaccine roll out. The same can be said of human psychological, sociological, and cultural resources. These last set of resources have not been emphasized in the national strategy. 

Thematically, the emphases and gaps highlight the segmented, selective, and siloed logic of the strategy, more so than in the case of India. While the guidelines cover most of the elements of the framework (and hence the system) as shown in the monads map (Figure 3) the coverage is not systematic. The primary theme is to use national government as a catalyst for the vaccine roll out. It is a simple but undifferentiated strategy—it does not differentiate by population, actions, resources, and levels. Since the beginning of vaccination development, the U.S. national government has used mass vaccination strategies that act as a catalyst to nudge the actions. Catalysts used by the U.S. national government such as funding for operations, removal of regulatory barriers, strengthening delivery infrastructure, and emergency use led to progress in scaling the vaccination process [39,41,42]. 

The primary theme misses out on the critical resources that can be used as a catalyst for different actions for population types. The issue of reaching underprivileged populations and reducing the distance to secure the vaccine has been a critical component. Research indicates that the accessibility of vaccination sites will have implications for equitable access to the vaccine, given that people of color and lower-income individuals are more likely to face location-based barriers to healthcare [43]. 

The secondary theme is to use temporal availability and informational stimulant as drivers to manage the vaccine roll out. This strategy is undifferentiated by population, stakeholder, and level. To ensure availability, the government had heavy investments both in Research and Development (R&D) and in manufacturing capacity. The manufacturing capacity enabled the production of large quantities of vaccine before the results of the Phase III trials were available. There was a chance that the government was potentially absorbing the full financial risks of R&D failure [41]. Though there was authorization of emergency use of the Pfizer vaccine at the initial stages and again for adolescents [44], sharing results of the trials with the public acted as a driver to manage mass vaccination. Although the theme covers the availability aspect, it does not cover the barriers for vaccination access to different population segments. As mentioned above, there are location-based barriers for people of color and lower-income individuals. In addition, there are availability barriers to these population segments in terms of accessing these resources [45]. Although there is importance given to informational stimulant, it has been to manage the vaccination program. It lacks targeting vaccination hesitancy by use of informational stimulants to gain the trust of the public. According to experts, in the African-American community, existing problems of mistrust create a vaccine hesitancy that did not necessarily exist before the pandemic or was less evident than now [45].

The tertiary theme indicates deploying spatial location and other human resources to mobilize the vaccine roll out but excludes the forces, levels, populations, and stakeholders. Involving different stakeholders and leveraging the health system infrastructure to its advantage, the U.S. strategy for vaccination involves location resources and other human resources for carrying out vaccination. The plan of the government has been to involve providers such as pharmacies and public health departments to have government-run vaccination sites to increase vaccine uptake [43]. On the other hand, while the strategies have focused on the different providers, the level at which the operations will be handled has less emphasis. The role of state and local government is not given enough emphasis, and this is likely to influence the distribution. The governments at all levels have to significantly expand their distribution channels and partnerships for vaccine administration to reach the target groups [43].

The quaternary theme is the least segmented, focusing on a variety of resources at the national level that can inhibit a wide range of actions by community and other agents in the vaccine roll out. It too excludes the consideration of population. There are issues related to the national-level informational educational inhibitors. One of the informational issues has been use of the COVID-19 vaccination campaign for financial gain. While this is not the only factor in determining adoption behavior, through effective communication these issues can be addressed and can build public confidence [46]. Strategies on transportation of vaccines has been another major focus due to the cold storage requirements. Complicated links of the supply chain must hold from the time of the truck pickups from the factories that head to airports and other distribution centers. Their maintenance requires coordinated activities [47]. With the strategies being laid out without full consideration of the target populations, it is not comprehensive because in the end the entire population is to be vaccinated. As vaccination drives progress, federal officials are looking to adopt different strategies such as expanding smaller plans and using mobile vaccination clinics for hard-to-reach populations and pushing education campaigns [48]. 

The quinary theme highlights significant systematic oversights in all dimensions. They are: (a) State, District, and Block levels; (b) spatial distance, financial income, and human psychological, sociological, and cultural resources; (c) barriers; (d) location as action; (e) population segments; and (f) most of the government, healthcare provider, and agent stakeholders. For example, people who work and get paid by the hour will lose out on the wages while being in line to get their vaccinations. Mechanisms previously employed in small-pox-vaccine drives in which people were reimbursed for medical care and received lost-income benefits could be adopted as a strategy to promote equitable distribution [10]. There are also barriers related to human psychological, sociological, and cultural resources. Anticipating and combating negative attributions from the mentioned resources requires building trust and addressing false attributions promptly and consistently.

### 4.3. Comparison of India’s COVID-19 Vaccines Operational Guidelines and the U.S. National Strategy for the COVID-19 Response

Both India and the U.S. place heavy emphasis on the resources and actions, and low emphasis on the population. The U.S. places greater emphasis on the forces than India, but both underplay the barriers and inhibitors as compared to the catalysts and drivers. The U.S. places greater emphasis on the stakeholders than India; India places greater emphasis on the level of intervention than the U.S. The coverage of both is systemic but with different emphases. In hindsight, the low population specificity and lack of emphasis on a positive approach emphasizing the catalysts and drivers may have diminished their effectiveness.

The national guidelines cover the full spectrum of levels from the national level to the block, with the greatest emphasis on the district (in India’s context) and state level (in the U.S. context). They are consonant with the federal structure of these countries wherein health is both a federal and state subject. The actions are both centralized and decentralized. They cover the full spectrum of actions except to locate the populations. They seem to presume the ease of locating the population for the vaccine roll out with no barriers or inhibitors. Recent experience highlights the difficulty of doing so. To overcome the barriers of locating different population segments in the U.S., Santa Clara County public health department assigned contact tracers as vaccine navigators to call people who have had a possible exposure to COVID-19 to tell them when they are eligible to get vaccinated [49]. While the guidelines cover a wide range of stakeholders, the low-to-no emphases on families and international government is an oversight. Both have emerged as important stakeholders for different reasons. The former for psychological, sociological, and cultural reasons; the latter because of the unexpected prominence of vaccine diplomacy. The need to utilize dissemination of information, as a potential driver, to communicate about vaccine development, its safety and efficacy, time needed for providing protection, and the importance of herd immunity is crucial [31].

Neither approach is systematic based on the themes maps. Both emphasize resources, forces, and actions, and both ignore the population. India’s approach, however, encompasses the levels of action and stakeholders comprehensively; the U.S. approach encompasses very selective levels and stakeholders.

Overall, the coverage of both the Indian and U.S. COVID-19 vaccine guidelines is not systematic or systemic. Oversights/gaps in terms of formulating inter-linkages within the elements of a given dimension and between the elements of the dimensions are evident. For instance, in the case of India, the primary theme to use informational stimulant as a driver to manage and redirect vaccine roll out is not linked to informational educational or any other elements mentioned in the resource dimensions. In the case of the U.S., use of catalysts for vaccine roll out by the national government is not linked to use of resources as catalysts by any other stakeholder. Likewise, the connections between elements of action with that of level, resource, force, population, and stakeholder dimensions have resulted in emphases being segmented, skewed, and not systemic in their approaches. Harnessing such systemic approaches is critical for systematic vaccine rollout and could aid in addressing contemporary issues.

## 5. Conclusions

The nature of the pandemic is such that vaccination is an integral tool to stop and mitigate the impact of the pandemic. Reports of diverse sources of information varying from scientific evidence to social media providing contradictory information, preference to natural immunity, misinformation regarding the vaccine, lack of perceived safety, fear of having side effects, and government conspiracies [31] have further challenged the drive for vaccine rollouts. While testing, tracing, tracking, and vaccination are prominent precursors for managing COVID-19, there is a need for formulating appropriate post-manufacturing surveillance to document the efficacy of the vaccines across different patient groups [50]. Such information could aid in effective feedback for learning. Planning ahead and devising alternatives similar to the global advances in home and community-based care have to be accelerated to enhance diagnosis, therapeutics, remote monitoring, and telemedicine [36] approaches. 

Overall, the strategies presented in the national strategy documents are guiding frameworks that evolve once the implementation starts. The ontology and the analysis of the present vaccine guidelines corpus can help comprehend the pathways that are emphasized, not emphasized, and absent during the national COVID-19 vaccine roll outs. For the strategies and guidelines to be effective, they must be systematically directed by a systemic framework. The present selective and segmented approach of practice in the domain needs to be systemically conceptualized. 

A systematic and systemic framework will provide feedback on and aids learning about the gaps in guidelines and the practice. The learning from feedback will reduce the gaps and make both guidelines and practice strategies more effective.

This paper contributes a framework and a method to formulate, implement, assess, and reformulate national vaccine rollout strategies systemically and systematically. It illustrates the application and value of the method through application to the strategies of India and the U.S., respectively. The framework can be adopted to analyze other strategic national guidelines of vaccine roll out, and the attendant analysis can be used to develop a roadmap for research on the subject. It can be used to integrate research with policies and practice, as discussed above. It can also be used to develop an agenda for research to fill the gaps in the state-of-the-research, -policies, and -practice.

## Figures and Tables

**Figure 1 ijerph-18-07483-f001:**
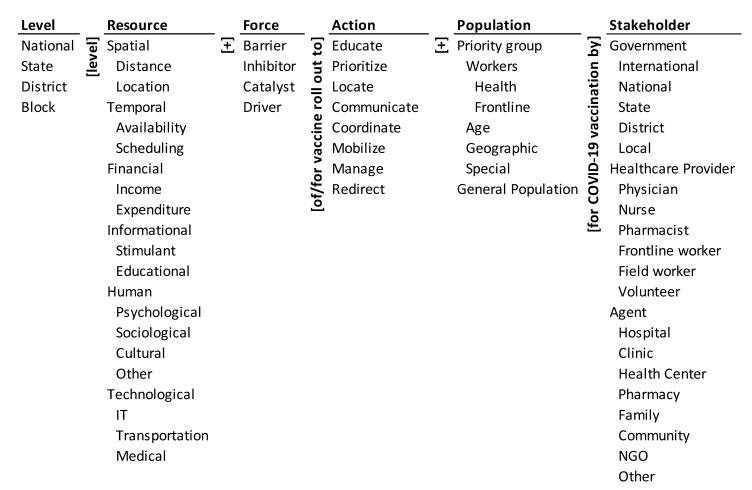
Ontology of COVID-19 vaccine roll out.

**Figure 2 ijerph-18-07483-f002:**
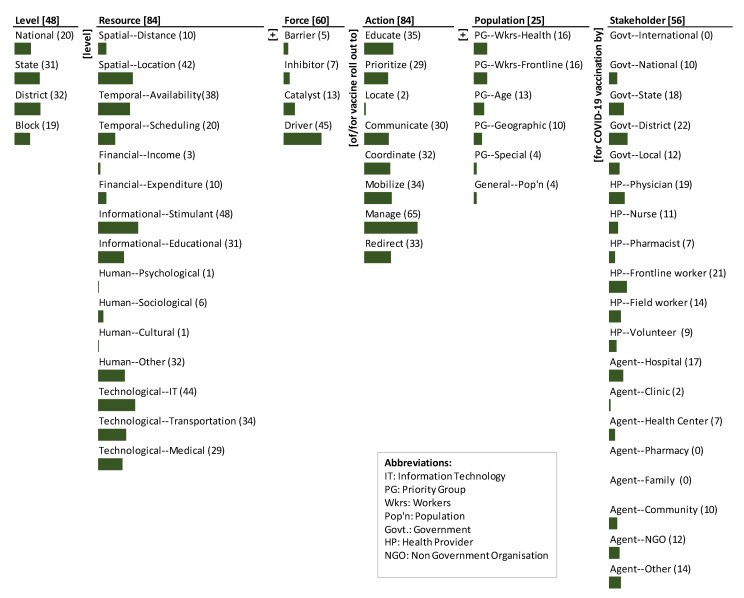
Monads map of India’s COVID-19 Vaccines Operational Guidelines.

**Figure 3 ijerph-18-07483-f003:**
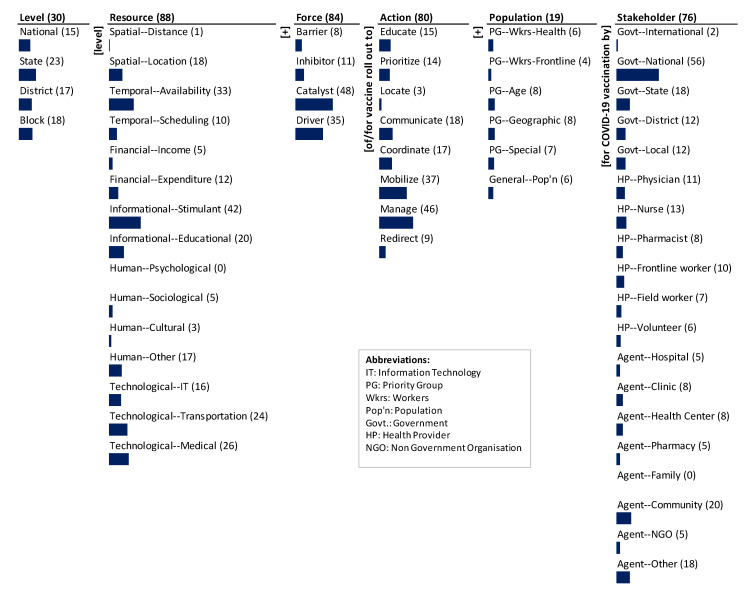
Monads map of the U.S. National Strategy for the COVID-19 Response and Pandemic Preparedness.

**Figure 4 ijerph-18-07483-f004:**
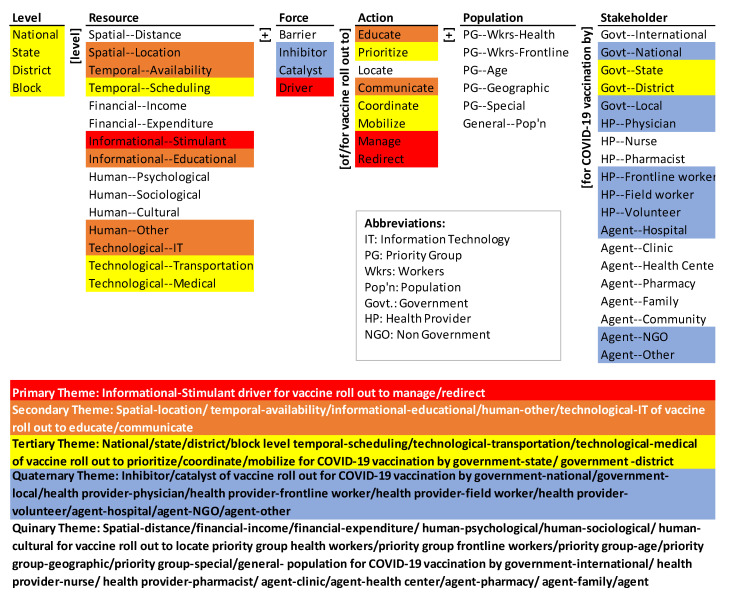
Themes map of India’s COVID-19 Vaccines Operational Guidelines.

**Figure 5 ijerph-18-07483-f005:**
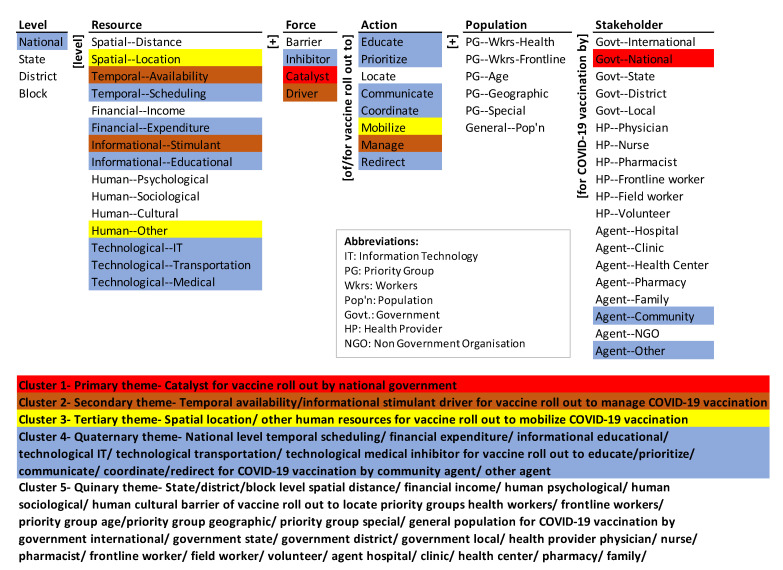
Themes Map of the U.S. National Strategy for the COVID-19 Response and Pandemic Preparedness.

**Table 1 ijerph-18-07483-t001:** Inclusion and exclusion criteria.

Name of the Document	Inclusion Criterion	Exclusion Criterion
COVID-19 Vaccines Operational Guidelines by the Government of India (GoI) National Strategy for the COVID-19 Response and Pandemic Preparedness by the office of President Joseph R. Biden, Jr.	Strategies/action points specifically pertaining to COVID-19 vaccine roll out	Strategies/action points not pertaining COVID-19 vaccine roll out. For example, Features of Co-WIN website (India’s document) was not considered for coding, since it was not directly related to vaccine roll out.

## Data Availability

The datasets used and/or analyzed during the current study are available from the corresponding author on reasonable request.

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
