# Peer review of "Ontological Analysis of COVID-19 Vaccine Roll out Strategies: A Comparison of India and the United States of America"

_ijerph, 2021, doi:10.3390/ijerph18147483_

Round 1

Reviewer 1 Report

The work entitled "Ontological analysis of COVID-19 vaccine roll-out strategies: A Comparison of India and the United States of America" provides an analysis of vaccination strategies in India and the US using a methodology capable of highlighting the gaps and the strengths of the different strategies reflected in the documents by means an ontological framework.
In general, the article is interesting and allows to compare the two documents of strategies. However, it is somewhat repetitive in its writing and is excessively long. The methods section is difficult to understand. I think some paragraphs would be better included in methods, for example, between lines 214 and 223. On the other hand, a large part of the results section could be omitted since they are dedicated to describing what is shown in figures 2, 3, 4, and 5.
There are very similar paragraphs, such as those between lines 239-248 and 263-271.
The discussion is also excessively long, making the message that the authors try to convey with their work is unclear.

Reviewer 2 Report

Dear Authors, 

in my opinion , the paper is interesting but it needs to be improved. In detail:

1) the introduction is clear but I suggest to better specify the choice of India and Usa.

2) the material and method section is understandable and well written.

3) The results and discussion sections are quite clear, maybe too descriptive. It could be important to "discuss" using a critical point of view.

4) The conclusion section can be improved. Is the framework useful in other countries? What's the difference between systematically and systemically? Can you better explain?

Finally, I suggest to use the impersonal form (third person) in the entire paper.

Round 2

Reviewer 1 Report

The authors have made changes in the manuscript and answered the review issues. The paper can be published now.